# Molecular and Cellular Insights into the Development of Uterine Fibroids

**DOI:** 10.3390/ijms22168483

**Published:** 2021-08-06

**Authors:** Alba Machado-Lopez, Carlos Simón, Aymara Mas

**Affiliations:** 1Igenomix Foundation, INCLIVA Health Research Institute, 46980 Paterna, Valencia, Spain; alba.machado@igenomix.com (A.M.-L.); carlos.simon@igenomix.com (C.S.); 2Department of Pediatrics, Obstetrics and Gynecology, Universidad de Valencia, 46010 Valencia, Spain; 3Department of Pediatrics, Obstetrics and Gynecology, BIDMC, Harvard University, Boston, MA 02215, USA

**Keywords:** uterine leiomyoma, steroid hormones, genetics/epigenetics, tumor-initiating cell, biomarkers, targetable pathways, tumor bulk/single-cells

## Abstract

Uterine leiomyomas represent the most common benign gynecologic tumor. These hormone-dependent smooth-muscle formations occur with an estimated prevalence of ~70% among women of reproductive age and cause symptoms including pain, abnormal uterine bleeding, infertility, and recurrent abortion. Despite the prevalence and public health impact of uterine leiomyomas, available treatments remain limited. Among the potential causes of leiomyomas, early hormonal exposure during periods of development may result in developmental reprogramming via epigenetic changes that persist in adulthood, leading to disease onset or progression. Recent developments in unbiased high-throughput sequencing technology enable powerful approaches to detect driver mutations, yielding new insights into the genomic instability of leiomyomas. Current data also suggest that each leiomyoma originates from the clonal expansion of a single transformed somatic stem cell of the myometrium. In this review, we propose an integrated cellular and molecular view of the origins of leiomyomas, as well as paradigm-shifting studies that will lead to better understanding and the future development of non-surgical treatments for these highly frequent tumors.

## 1. Introduction

Uterine leiomyomas (uLM), also known as fibroids or uterine myomas, are the most important benign neoplastic threat to women’s health, with an estimated lifetime incidence of up to 70% [1,2,3]. Clinically, they are the most common cause of hysterectomy and a major source of infertility and abnormal uterine bleeding; thus, this condition significantly affects patients’ quality of life, as well as exerting significant economic impacts on healthcare systems worldwide [4,5,6].

Although current medical approaches to the diagnosis of uLM are based mainly on imaging and histological assumptions, molecular tools are gaining relevance as an alternative to conventional strategies in all clinical fields [7,8,9]. Likewise, considerable progress in understanding the origin and development of this highly prevalent condition has been made in the last decade. In this regard, hormonal processes, genetic predisposition, somatic alterations (point mutations and chromosomal abnormalities), epigenetic disruptions and the cellular origin of uLM will be discussed in this comprehensive review to share up-to-date knowledge and new concepts regarding the basic molecular biology and pathophysiology of this condition.

Recent advances related to the identification of biomarkers for early and differential diagnosis of uLM that support current histology-based classification will be discussed, as well as the detection of targetable pathways to develop novel and improved options for clinical management of these benign tumors.

## 2. Uterine Leiomyoma Etiopathology

### 2.1. Hormone Features

Despite the extensive clinical evidence regarding the hormonal influences on uLM development, the functional role of estrogens and progesterone remains unclear [10,11,12,13,14]. Early menarche age is associated with a higher risk of developing uLM due to the longer duration of exposure to estradiol and progesterone, which are reduced during menopause [15,16,17,18]. Conversely, increased parity seems to reduce the risk of fibroid development, while nulliparity is related to higher risk of uLM. This situation may seem ambiguous since there are high levels of circulating estrogens and progesterone of placental origin; however, this paradox may be due to differentiation of the myometrium during pregnancy [19]. This process makes the tissue less susceptible to the action of growth factors [20] and genetic mutations that trigger uLM tumors. Furthermore, uLM present before a pregnancy are reduced in volume after the gestation, suggesting apoptosis induced by postpartum remodeling and ischemia during delivery [21].

Estrogen, together with estrogen receptor-a (ERa), renders leiomyomas uLM responsive to progesterone by inducing progesterone receptor (PR) expression (Figure 1). PR binds to tens of thousands of DNA sites in leiomyoma smooth muscle cells to regulate multiple genes and promote proliferation, survival, and abnormal production of extracellular matrix. This occurs mainly during the reproductive years, while low hormone levels due to menopause or GnRH analogue-therapies are responsible for tumor degeneration [22,23].

At the tissue level, high expression levels of progesterone receptor in the myometrium are also associated with increased risk of developing uLM [24], while progesterone is essential for the growth and maintenance of uLM [25]. Several mechanisms have been proposed for the relevance of progesterone in uLM, including that it induces the expression of proliferation genes such as the *BCL-2* family [26,27] and induces specific pathways such as the AKT pathway [28] or the MEK1/2 Rho/Rock pathways in response to mechanical signaling [29]. In fact, progesterone receptors are one of the most common therapeutic targets in uLM management due to the use of selective progesterone receptor modulators, including ulipristal acetate (UPA) or mifepristone, that can inhibit proliferation, increase apoptosis, and reduce tumor growth and symptoms [30,31].

The interaction between progesterone and estrogens, their respective receptors, and other paracrine signals is also key in the etiopathogenesis of uLM. Progesterone may suppress estrogen receptors [32], but estrogen disruptions can alter the function of both estrogen and progesterone-associated genes and pathways [33]. Estradiol, the main form of estrogen, can trigger production of specific growth factors, mainly *PDGF*, through the MAPK-PKC pathway, leading to increased proliferation or immortalization [34,35,36]. Estradiol can also activate the Wnt/β-Catenin pathway through ERα to promote proliferation [37]. Similar to progesterone, inhibition of estrogen activity has been proposed, mainly by using aromatase inhibitors [38].

These findings show that progesterone and estrogen play a key role in uLM pathogenesis; however, further research on their involvement is needed to explore new therapeutic approaches that exploit the specific mechanisms of action of these ovarian steroid hormones. These novel fibroid therapies hold immense promise for shifting mainstream treatment of uLM from the surgical domain to the realm of orally administered medicines.

### 2.2. Genetics

Like most tumors, uLM have complex genetic backgrounds, including germline alterations and somatic mutations (point mutations and chromosomal abnormalities). These changes may be both the causes and consequences of the mechanisms behind the development of these uterine tumors (Figure 2).

#### 2.2.1. Inherited Susceptibility

Although leiomyomas frequently appear spontaneously, there is a heritable component in their development with heritability values ranging from 8% to 70% [39,40,41]. Some of the first works that attributed some degree of heritability to leiomyoma development were twin studies that observed higher concordance levels between monozygotic twins than between dizygotic twins [39,42].

Family-based association studies have also found that the incidence of leiomyomas was higher among first-degree related women than among unrelated individuals [43,44,45]. Further, familial prevalence of fibroids was significantly associated with higher frequency of specific symptomatology [46]. One of the most relevant examples of familial association in uLM is hereditary leiomyomatosis and renal cell carcinoma (HLRCC), an autosomal dominant cancer syndrome characterized by skin leiomyomas, uterine fibroids, and kidney tumors [47]. HLRCC is caused by germline mutations in the *fumarate hydratase* (*FH)* gene, which encodes an enzyme involved in the tricarboxylic acid cycle with a tumor suppressor role [48,49]. In HLRCC, biallelic inactivation of *FH* causes deficient oxidative phosphorylation, which can produce the high levels of ATP required for rapid cell proliferation in tumoral cells through the Warburg effect [50].

Lastly, genome-wide association studies (GWAS) have sought genetic variants that may predispose women to uLM. Some studies found distinct susceptibility loci depending on the race/ethnicity of the patients, which explains the differences in incidence rates between populations such as the increased risk of uLM in women of African ancestry [51,52,53,54].

Although it is challenging to find specific genes involved in uLM development due to these differences in susceptibility loci between races/ethnicities, specific associations have been found between genes, and not only related to uLM risk but also uLM size [55,56]. So far, reported genes associated with uLM include *ODFC3, BET1L, RIC8A, SIRT3, SLK, OBFC1, TNRC6B, FASN* and *HMGA2*; however, the specific mechanisms underlying the associations remain unclear [57]. One GWAS also associated candidate genes for age of menarche to uLM, suggesting a possible explanation for increased uLM risk in women with early menarche age [58].

#### 2.2.2. Point Mutations 

Besides germline mutations that may predispose individuals to uLM, these tumors also accumulate several somatic mutations that usually affect a widely researched subset of genes and genomic regions. One of the most relevant genes affected by somatic mutations is *MED12,* which is mutated in up to 70% of uLM [59,60,61,62].

In normal physiological conditions, *MED12* belongs to the mediator kinase module, which is a regulator of RNA polymerase II-mediated transcription. Binding of *MED12* activates the catalytic core of this module, *CKD8*, by placing an “activation helix” and preventing the binding of kinase inhibitors, which allows precise transcription regulation [63]. However, *MED12* mutations or altered expression are widely reported in many human diseases, including behavioral disorders and several cancer types such as breast, prostate, colon, ovarian and lung cancer [64].

In uLM specifically, *MED12* mutations frequently occur in codon 44 of exon 2 [65] and less frequently in exon 1 [66], usually in the absence of any other recurrent mutations (Figure 2). This finding indicates that *MED12* alterations alone may be sufficient to cause tumor development [65]. Although less frequent, a hotspot for small deletions in *MED12* has also been detected in uLM, which may be caused by non-canonical DNA structures located in this hotspot [67].

One of the driver mechanisms behind *MED12* mutations is disruption of the *MED12–CDK8* interaction, which can lead to altered expression profiles of many genes due to altered transcription regulation [61]. Further, gain-of-function mutations in *MED12* may cause genomic instability [68], possibly by inducing the Wnt4/β-Catenin pathway, which may induce cell proliferation, tumorigenicity and impaired autophagy through mTOR signaling [69,70].

In addition to its role as a driver gene, *MED12* alterations are associated with clinical features of uLM including tumor size, conventional histology, and subserosal location as well as diagnosis of multiple vs. single uLM [71,72]. Moreover, the number of MED12-positive fibroids is inversely related to parity, whereas the number of mutation-negative tumors is positively associated with a history of pelvic inflammatory disease [73].

Other genes are reported to accumulate specific point mutations in uLM, including genes involved in the cell cycle and tumor suppression such as *CAPRIN1*, *DCN*, and *AHR* [74] as well as specific mitochondrial genes [75].

#### 2.2.3. Chromosomal Abnormalities

uLM also display alterations that affect whole genes or larger chromosomal regions (Figure 2). Aberrations are found in several different chromosomes, primarily chromosomes 7, 12, 14, and 15 [67,76,77,78]. Such cytogenetic alterations are proposed as criteria to divide uLM into subgroups with different molecular and clinical features [79]. In general, tumor size is increased in cytogenetically abnormal fibroids [80,81], while deletions in specific regions such as chromosome 1p may be associated with histopathological variants of uLM and with prognosis [82].

Chromothripsis

Chromosomal rearrangements in uLM sometimes present as numerous deletions with many breakpoints affecting limited genomic regions in only one of the homologous chromosomes, a process called chromothripsis [83]. The causes behind this type of complex chromosomal rearrangement remain unclear, although different mechanisms, such as chromosome pulverization, abortion of apoptosis, and telomere shortening, are proposed as drivers in different cancer types [84].

In uLM, chromothripsis appears to be more moderate than in other cancer types (Figure 2), which is why it is also referred to as a chromothripsis-like phenomenon [85]. This type of event is not associated with specific *MED12* mutations or other alterations and is limited to up to 4 chromosomes, although the proportion of uLM presenting chromothripsis-like profiles varies from 20% to 42% [86,87,88]. As expected, the specific chromosomes affected by this phenomenon also vary between studies since chromothripsis occurs due to apparently random processes that cause breakages in different chromosomes. Although chromothripsis was initially associated with malignant processes, the detection of rearrangements in numerous genomic regions in uLM indicate that it may be one of the mechanisms in the etiopathology of this tumor [89].

Target genes of chromosomal alterations

While chromosomal aberrations can happen anywhere in the genome in uLM, specific genes are frequently affected by these types of events, leading to disrupted biological processes that may drive tumorigenesis (Figure 2).

Rearrangements in the 12q14–15 region in fibroids are reported to increase expression of the *HMGA2* gene [90,91]. Altered expression of this protein, which is involved in transcription regulation, has been associated with leiomyoma development through different mechanisms including angiogenesis [92], ERα-mediated cell proliferation [93], and homologous recombination DNA repair, since one of the preferential translocation partners of *HMGA2* is the DNA repair gene *RAD51B* [94,95]. Another member of the family, *HMGA1*, is also affected in uLM by rearrangements targeting region 6p21. HMGA1 and HMGA2 may have a similar physiological role, which explains why disruptions of both genes result in similar consequences [96,97].

Conversely, while deletions in the 7q22 region have been associated with uLM [98,99], the affected target gene has not been conclusively identified. While these deletions alter expression of several genes such as *LHFPL3, LAMB1,* or *HPB1* [100,101,102], one of the most accepted target genes is *CUX1.* 7q deletions can result in both increased and reduced expression of CUX1, suggesting a tumor suppressor or oncogenic role [103,104] that may cause uLM development in a process similar to *RAD51B* [105].

Lastly, deletions in the X chromosome affecting the collagen IV genes *COL4A5* and *COL4A6* associated with Alport syndrome have also been associated with development of smooth muscle tumors (diffuse leiomyomatosis), including uterine leiomyomas [106,107,108].

### 2.3. Epigenetics

In humans, epigenetic mechanisms, which affect gene expression, play a crucial role in leiomyoma formation [109,110]. The main epigenetic mechanisms include regulation by DNA methylation, histone modifications, microRNAs (miRNAs), and long-noncoding RNAs (lncRNAs) (Figure 2).

DNA methylation is essential for normal development and aberrations in this process affecting key embryonic development genes such as *FOXO1*, *TERT*, and *WNT4* [111], which are associated with hypermethylation of tumor suppressor genes and/or hypomethylation of oncogenes, could contribute to tumorigenesis [112,113]. Specifically, uLM are associated with alterations of DNA methylation, increased mRNA expression of estrogen receptor 1 (*ESR1*; [114]), and DNA methyltransferases in tumor samples compared to the normal myometrium [115,116,117,118]. Recently, it has also been demonstrated that upregulation of HMGA2 expression does not always appear to be dependent on translocation but is associated with hypomethylation in the *HMGA2* gene [119]. Additionally, DNA methylation and *MED12* mutation together constitute a complex regulatory network that affects progesterone/PR-mediated *RANKL* gene expression [120,121], with an important role in activating stem cell proliferation and leiomyoma development [121]. Together, these findings suggest that DNA methylation might play a key role in the pathogenesis of uterine leiomyoma by altering the normal myometrial mRNA expression profile. Further studies involving epigenetic modulators such as 5′-Aza-Cytidine could lead to novel treatments that slow tumor growth by depleting the stem cells through specific demethylation mechanisms [93,122].

Other epigenetic alterations include methylation and acetylation of the histone tails, with enhancer of zeste homolog 2 (EZH2) and histone deacetylases (HDACs) being the main enzymes involved in these processes related to uterine leiomyomas [109]. While histone methylation can determine either activation or repression of gene transcription, histone acetylation only regulates gene activation [123,124]. Particularly, in uLM, EZH2 methylation silences gene function [125,126], though HDACs are involved in the regulation of tumor suppressor gene *KLF11*, which is diminished in these benign tumors [127].

Lastly, it is important to note the significant role of microRNAs from the let7, miR-21, miR-93, miR-106b, and miR-200 families in the epigenetic mechanisms contributing to the proliferation, inflammation, angiogenesis, and synthesis of extracellular matrix components. Thus, they contribute to uLM development through relevant signaling pathways such as Wnt/β-catenin and Wnt/MAPK [128,129,130,131,132]. Recently, expression of lncRNAs (RNA transcripts of more than 200 nucleotides) such as XIST was reported to be altered in uLM leiomyomas compared to myometrium [133].

Advancing our understanding of the role of epigenetic regulations in the risk of uLM and their contribution to tumorigenesis (probably associated with a hyperestrogenized phenotype during myometrial development) will help to identify novel therapeutic options for uterine leiomyoma [134].

### 2.4. Cell Origin

Considering that uLM are monoclonal tumors [135], it is possible that dysregulation of committed cells that acquire stem-like features could be responsible for this benign condition (Figure 3).

Indeed, cells with stem or progenitor characteristics can be isolated from myometrial and leiomyoma tissues [136,137]. Interestingly, these cells seem to have low levels of sex steroid hormone receptors but require these steroids for tumor growth, suggesting that an additional paracrine mechanism, exerted by mature surrounding cells that express these receptors, is necessary for the transformation of these cells [138]. In fact, early-life exposure to endocrine-disrupting chemicals such as xenoestrogens can alter the characteristics of progenitor cells, causing them to become tumor cells [139].

Moreover, progenitor cells from uLM, but not from the myometrium, carry *MED12* mutations, indicating that at least one genetic hit may transform a myometrial stem cell into a tumor-initiating cell and give rise to these benign tumors [140,141]. Mutations in myometrial cells that affect the expression of the *HMGA2* gene are also reported, leading to the formation of leiomyoma-like tissue in xenograft models [142]. Although alterations in *MED12* and *HMGA2* seem to be mutually exclusive [143], whether these genetic alterations induce the transformation of myometrial stem cells or maintain already existing leiomyoma stem-progenitor cells remains unclear.

Ultimately, other environmental and epigenetic conditions such as uterine hypoxia or aberrant methylation could play a critical role in leiomyoma development and growth from the cellular point of view [93,144,145].

In summary, while further characterization of the cellular origin of uLM is needed, these myometrial stem/progenitor cells offer novel and powerful potential targets for therapeutic or preventative strategies. Because transformation of myometrial stem cells into pre-fibroids seems to be a widespread if not ubiquitous process, future interventions will probably target the growth acceleration phase of fibroid development.

## 3. Molecular Tools in Clinical Management of Uterine Leiomyomas

Given the relevance of molecular processes in the development of uterine leiomyomas, there is a broad spectrum of molecular tools and resources that could be employed to develop better detection, classification, and treatment methods (Table 1). These promising approaches may help to improve the outcome and quality of life of symptomatologic patients and reduce costs and possible complications.

### 3.1. Detection–Putative Biomarkers

Clinically, tumor biomarkers could be useful for the identification of women at higher risk of developing fast growing leiomyomas as well as to distinguish this benign condition from other pathological conditions. Several biomarkers show increased serum levels in women with uLM compared to women without uLM, including leptin, VEGF, ghrelin, and CA125 [146,147,148]. However, only some of these show significant differences in healthy and fibroid patients and could be used in clinical practice. Potential serum biomarkers for uLM include proteins involved in oxidative stress, such as vitamin D3 or AOPP [149,150,151,152], inflammatory processes, such as YKL-40 [153] and TRADD [154], and angiogenesis, such as TGF-β and LRG1 [155].

Further, biomarkers could be used to distinguish uLM from other uterine malignancies to improve prognosis and treatment, for instance, in deciding whether the uterus needs to be completely resected. Differential diagnosis of uLM and uterine sarcomas could be based on GDF-15 [156], and CA125 [157] serum levels, although their applicability is limited since increased expression of these biomarkers is more typical of high-grade tumors. Similarly, HLA-G levels differ between patients with uLM and patients with ovarian endometriosis, although these levels are greatly dependent on the phase of the menstrual cycle, which limits their clinical potential [158]. Although many different biomarkers of uLM have been reported, still none of them are currently used in clinical practice. This highlights the need for further research and validation of these tools and their clinical applicability.

### 3.2. Classification–Gene Expression Profiling

Although classification of uLM is based mainly on their localization and histopathological features, efforts are focused on developing a more precise new classification system that may aid diagnostic and therapeutic decision making.

Molecular classification of uLM has, so far, relied on dividing them into different categories based on their mutational status for the main “driver genes” (*MED12, FH, HMGA2),* which are usually mutually exclusive [159,160]. However, some uLM do not fall within any of these categories, or could belong to more than one category, for example, by presenting both *MED12* and *HMGA2* alterations [161,162,163]. This demonstrates that further delineation of the features of these subtypes and an improved classification system are necessary.

Gene expression profiling could be used not only to unravel the specific mechanisms behind the pathogenesis of the main molecular subtypes of uLM, but also to define new subgroups based on their transcriptomic signatures. Transcriptomic analysis clustered uLM samples based on their driver mutations [95], although functional analysis of dysregulated genes showed that the involved cellular processes were in fact very similar in molecular subtypes, e.g., mitosis, cell cycle, and angiogenesis [119].

Lastly, transcriptomic analysis of tissue biopsies could also be applied to the differential diagnosis of myometrial tumors to accurately distinguish leiomyomas from leiomyosarcomas [164,165].

### 3.3. Treatment–Targetable Dysregulated Pathways

Recent omic approaches have generated large amounts of information about the ethiopathology of uLM. This opens a new field that may allow clinicians to choose better treatments based on the specific molecular pathways that are altered in each uLM. Although many different pathways are altered in uLM, in this review we will focus on three we have found to be most relevant from a therapeutic point of view.

#### 3.3.1. Wnt/β-Catenin Pathway

Disruptions in Wnt/β-catenin signaling, a conserved pathway that regulates proliferation, migration, and tissue homeostasis among other processes, have been reported in different cancer types, including breast and gynecological cancers such as ovarian cancer [166]. In uLM, activation of this pathway stimulates proliferation and stem cell function [167], while inhibiting it can lead to reduced cell growth [132]. Besides, environmental conditions such as hypoxia or serum starvation may inhibit this pathway in cultured human leiomyoma cells, and induce their transdifferentiation, leading to the development of lipoleiomyomas, a less prevalent variant of uLM with a high content in adipocytes [168].

The importance of this pathway in the development and progression of uLM makes it of special therapeutic interest. In fact, vitamin D3 has already been proposed as a potential Wnt4/AKT/β-catenin inhibitor that could be used to reduce leiomyoma growth and proliferation [169,170].

#### 3.3.2. PI3K/AKT Pathway

The PI3K/AKT pathway is also frequently altered in different cancer types, where disruptions in the function of phosphatidylinositol-3 kinases, AKT, or mTOR can result in increased or uncontrolled cell survival and growth [171].

In uLM, over-activation of this pathway or its components are associated with growth and development through different genes such as *GSK3*, *PTEN*, *AKT*, *CCND2*, and *P27* [172,173,174,175]. This pathway is inhibited by widely used therapeutic options for uLM such as GnRH analogues [176] and ulipristal acetate [169].

#### 3.3.3. TGF-β Pathway

Transforming growth factor β is one of the most relevant cytokines involved in the proliferation and differentiation of myometrial tissue [177]. Overexpression of this cytokine and alteration of the TGF-β pathway were reported in different studies [178,179,180,181]. Accordingly, therapies that decrease TGF-β signaling, including GnRH analogues [182] and ulipristal acetate [183,184], succeed in reducing uLM growth and symptomatology.

## 4. Clinical Translation and Future Research

Recent molecular studies based on genomic and transcriptomic approaches have revealed driver mutations, aberrant regulatory programs, and disease subtypes for major human myometrial tumors [119,165,185]. However, these studies were focused on the whole tumor mass, which implies certain limitations due to tumor heterogeneity, which can hide critical differences between cells within populations [186,187]. In this sense, intratumoral heterogeneity represents one of the greatest challenges in tumor biology and oncology. In fact, interactions between cells, as well as local and autonomous mechanisms within the tumor microenvironment could be essential for better understanding the origin and evolution of uterine fibroids. In addition, the local and autonomous mechanisms of cells could play a key role in tumor development and growth.

To further examine the function and regulation of these cell populations, unbiased genome-wide studies of their gene expression are needed. In the last decade, the exponential growth of single-cell transcriptomic technologies enables processing hundreds of thousands of cells at an affordable cost and time [188,189,190,191]. One of these technologies, “Smart-seq2”, can generate quantitative and reproducible data since it has improved reverse transcription and pre-amplification to increase the throughput and the length of cDNA libraries generated from single cells [188]. Recently, standardization of droplet genomic methods democratized this technology, making it possible to reduce the batch effects when integrating data obtained from different geographic locations. These developments led to the first atlases of human tissues, which have provided significant information about multiple biological systems [192,193,194]. Full characterization of the individual cell types present in human myometrium and leiomyoma would help to clarify the fundamental physiological and pathological processes involved in uLM and further improve disease-related diagnosis and treatment.

Complementary approaches based on targeted nanoparticles could also have promising applications for the detection and therapy of this benign condition [128,195]. Modern studies have assessed nanomaterial-mediated delivery of anti-tumor cytokines [196] or adenovirus [197]. This delivery method results in increased apoptosis and suppressed tumor proliferation compared with other medical treatments. Thus, it could be interesting to consider the use of dedicated systems to deliver specific drugs for targeted non-invasive treatment of this pathology instead of surgery.

Finally, CRISPR/Cas9-mediated silencing of the AP-1 subunits in human uterine smooth muscle cells revealed that the loss of AP-1 can trigger large-scale changes in gene expression and acetylation, which could explain the wide transcriptional changes seen in leiomyoma pathogenesis [198,199]. Consequently, the CRISPR/Cas9 system could be used to correct specific mutations. In fact, in the upcoming decade, genome editing tools could help to accelerate therapeutic directions that specifically target the molecular mechanisms of uLM.

In summary, these and other novel techniques will be the next steps toward a better understanding of uLM from a research and clinical perspective (Figure 4).

## 5. Conclusions

Despite being benign tumors of myometrial origin, uterine leiomyomas have a considerable impact on women’s quality of life and on healthcare systems. Research in this field is growing rapidly, and the increasing novel molecular and cellular insights regarding this condition will allow scientists and clinicians to better understand the etiopathology of uLM.

These tumors are greatly influenced by levels of sex hormones that may trigger specific changes in an already altered genetic, transcriptomic, and proteomic environment. Uterine leiomyomas display great genetic complexity, ranging from germinal mutations that predispose women to developing uLM, to somatic alterations such as point mutations in the *MED12* gene or copy number variants and chromosomal aberrations that affect different regions, to epigenetic alterations, which complicates the utilization and interpretation of multi-omic information related to these tumors. Additionally, although it is widely accepted that uLM have monoclonal origins, the processes behind the transformation of myometrial stem cells into tumor-initiating cells remain elusive, further complicating our comprehension of uLM etiopathology.

In brief, understanding all the mechanisms behind uLM through the integration of differential cellular/molecular characteristics could facilitate the discovery of tumor-specific biomarkers, as well as the development of a more suitable uLM subclassification system, and therapeutic approaches specifically targeting the affected pathways. Application of current novel techniques and procedures such as single-cell sequencing or CRISPR-Cas genome editing to uLM research will further our understanding of the tumorigenic processes and improve the clinical management of this benign condition.

## Figures and Tables

**Figure 1 ijms-22-08483-f001:**
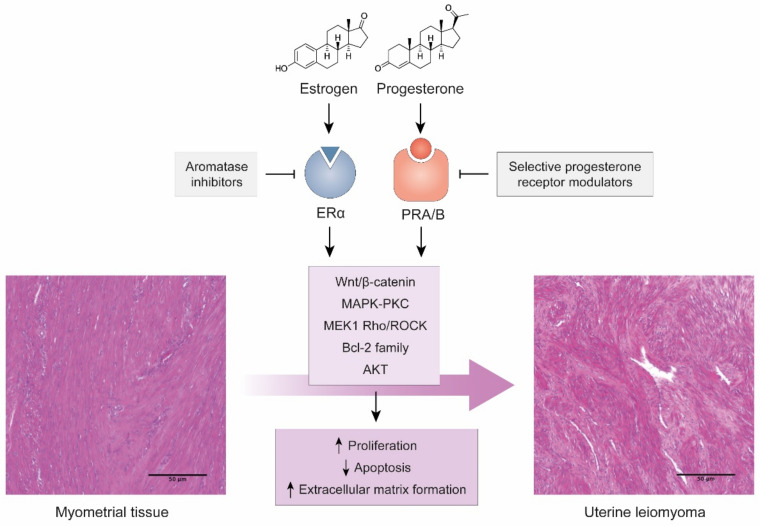
Hormonal action on myometrial tissue as a cause of uLM. Images of H&E-stained tissues from our group’s archives. Scale bars represent 100 µm.

**Figure 2 ijms-22-08483-f002:**
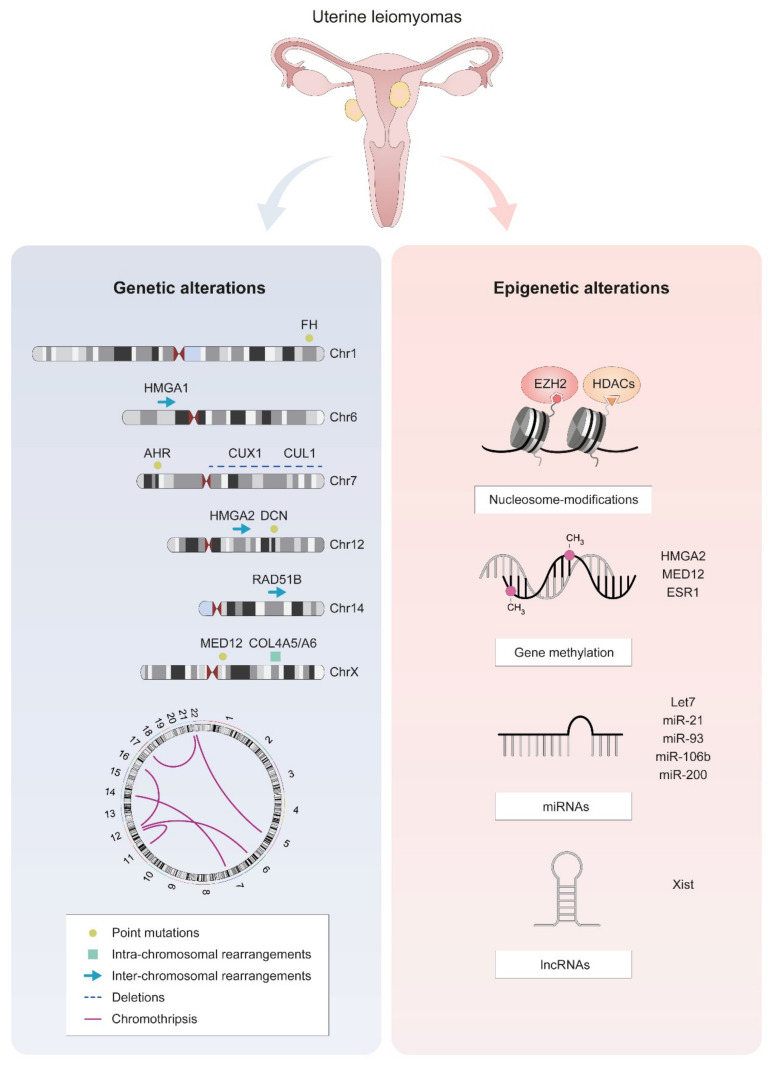
The most frequent molecular alterations in uLM cells. Genetic alterations are represented on ideograms of the most frequently affected chromosomes, along with a scheme of *HMGA1/2-RAD51B* and a circos plot representing chromothripsis. Epigenetic alterations include disrupted nucleosome modification, gene methylation, and altered expression of non-coding RNAs.

**Figure 3 ijms-22-08483-f003:**
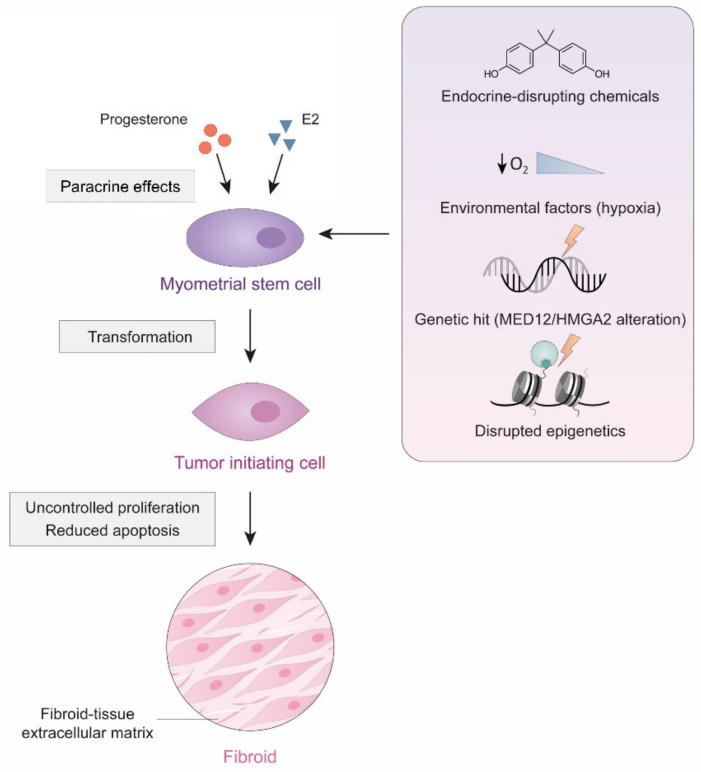
Cellular origin of uLM. Specific environmental and genetic changes may cause the transformation of myometrial stem cells into tumor initiating cells, that, through uncontrolled proliferation and reduced apoptosis, may eventually give rise to a uLM tumor.

**Figure 4 ijms-22-08483-f004:**
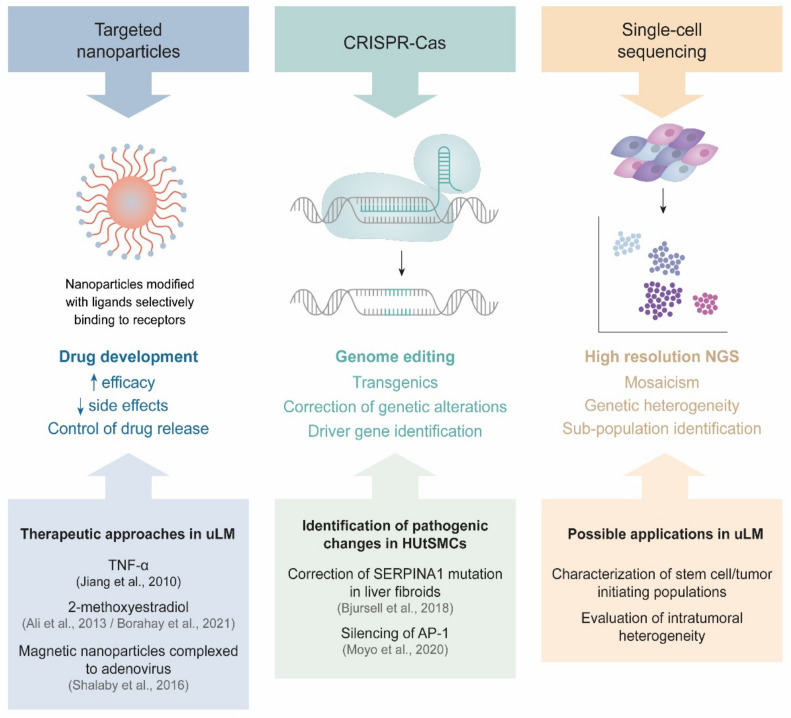
New techniques in uLM research. Each technique and its features are displayed, with relevant publications or future directions in uLM research.

**Table 1 ijms-22-08483-t001:** Molecular tools in clinical management of uterine leiomyomas. Possible applications include discovery of biomarkers for detection, classification based on gene expression profiling or pathway-targeting for treatment.

**Molecular Tools in the Clinical Management of uLM**	Detection: Putative biomarkers	↑ serum levels in uLM			Biomarkers for differential diagnosis		
		• Leptin	• CA125	• AOPP		• GDF-15	
		• VEGF	• Vitamin D3	• LRG1		• CA125	
		• Ghrelin	• TGF-β	• YKL40		• HLA-G	
	Classification: Gene expression profiling	Molecular subclassification			Differential diagnosis		
		Mutational status of driver genes			uLM versus uterine leiomyosarcoma		
		Similar cellular processes involved			Clearly distinct transcriptomic profiles		
		• MED12			High number of genes involved		
		• HMGA2			• BRCA2	• FOXM1	• PAX3
	Treatment: Targetable pathways	WNT-β catenin pathway		PI3K-AKT pathway		TGFβ pathway	
		↑ proliferation		↑ growth		Overexpression of cytokines	
		↑ stem cell function		Involved in development		↑ growth	
		Inhibited by vitamin D3		Inhibited by GnRH analogues and UPA		Inhibited by GnRH analogues and UPA	

## Data Availability

Not applicable.

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
