# Peer review of "Molecular and Cellular Insights into the Development of Uterine Fibroids"

_ijms, 2021, doi:10.3390/ijms22168483_

Round 1
Reviewer 1 Report
The article by Machado-López et al. reviews the molecular and cellular aspects involved in the development of uterine fibroids in the light of the most recent knowledge. The matter is relevant in a research and clinical perspective. The article is very well written in all its sections, brings many new informations and discusses and justifies potentially innovative clinical diagnostic and therapeutic strategies in this field.
I have only minor questions:
- Line 186: I suggest to check the sentence “Target genes of chromosomal alterations”, which appears to be unrelated to the context in which it is included.
- Tab. 1: Change “Grhelin” into “Ghrelin”
- Tab. 1: uLMS: the significance of this shortening should be clarified. I could not find uLMS in the text
- Lines 309-16: even though the perspectives reported in these sentences are fascinating, a little more caution than that reported in the text should be used. Currently, there is no large clinical application of these biomarkers in deciding the differential diagnosis between uLM and sarcoma outside academic research setting.
Author Response
Response to Reviewer 1
The article by Machado-López et al. reviews the molecular and cellular aspects involved in the development of uterine fibroids in the light of the most recent knowledge. The matter is relevant in a research and clinical perspective. The article is very well written in all its sections, brings many new informations and discusses and justifies potentially innovative clinical diagnostic and therapeutic strategies in this field.
We appreciate the reviewer’s opinion and positive feedback, as well as the proposed changes to improve the quality of our manuscript.
Point 1: Line 186: I suggest checking the sentence “Target genes of chromosomal alterations”, which appears to be unrelated to the context in which it is included.

Response 1: As the reviewer mentions, the sentence was unrelated to the context, since it was intended to be the title of the next section, discussing genes that are frequently affected by chromosomal alterations in uLM. The sentence has been moved to the following line, being the title of the next subsection (page 5, line 185).
Point 2: Tab. 1: Change “Grhelin” into “Ghrelin”
Response 2: Thank you for pointing out this typing mistake, we must have missed it during revision. The change has been made in Table 1, as suggested.
Point 3: Tab. 1: uLMS: the significance of this shortening should be clarified. I could not find uLMS in the text.
Response 3: The acronym “uLMS”, used to refer to "uterine leiomyosarcoma", has been changed in Table 1.
Point 4: Lines 309-16: even though the perspectives reported in these sentences are fascinating, a little more caution than that reported in the text should be used. Currently, there is no large clinical application of these biomarkers in deciding the differential diagnosis between uLM and sarcoma outside academic research setting.
Response 4: We agree with the reviewer’s opinion. Despite the differential diagnosis of uLM at the molecular level may be a very promising field, nowadays it is limited to research. To support how these biomarkers could have great clinical potential, we have added a small clarification (page 10, lines 317-320):
“Although many different biomarkers of uLM have been reported, still none of them are currently used in the clinical practice, highlighting the need for further research and validation of these tools and their clinical applicability.”
Reviewer 2 Report
The review article entitled "Molecular and cellular insights into uterine fibroids development" by Machado-López et al. is well organized to learn uterine leiomyoma. This reviewer has no major comment on the manuscript.
Author Response
Response to Reviewer 2
The review article entitled "Molecular and cellular insights into uterine fibroids development" by Machado-López et al. is well organized to learn uterine leiomyoma. This reviewer has no major comment on the manuscript
We really appreciate the reviewer’s opinion and thank them for taking the time to review our manuscript.
Reviewer 3 Report
The authors reviewed cellular and molecular views regarding the pathogenesis, development, and origins of uterine leiomyomas. The content is based on appropriate and adequate references. The manuscript is well written and reviewed.
Minor comments
#. Lines 257-265: Do the authors think that the progenitor cells have ER/PR expressions (or not)? Figure 3 seems to show that the progenitor cells possess ER/PR receptors, while the authors described the cells might have low-to-absent levels of ER/PR receptors. This is a very interesting point in this field. The reviewer thinks that the authors ' ideas can be shown.
Author Response
Response to Reviewer 3
The authors reviewed cellular and molecular views regarding the pathogenesis, development, and origins of uterine leiomyomas. The content is based on appropriate and adequate references. The manuscript is well written and reviewed.
We thank the reviewer for their opinion and positive feedback, as well as for taking the time to review our manuscript.
Point 1: Lines 257-265: Do the authors think that the progenitor cells have ER/PR expressions (or not)? Figure 3 seems to show that the progenitor cells possess ER/PR receptors, while the authors described the cells might have low-to-absent levels of ER/PR receptors. This is a very interesting point in this field. The reviewer thinks that the authors ' ideas can be shown.
Response 1: From our point of view, even though the expression of ER/PR in myometrial stem cells might be reduced or deficient, there is still some residual expression, which could be involved in their transformation and the development of uLM. However, the main source of E2/progesterone signalling in these cells are paracrine effects of surrounding mature/differentiated cells that do express high levels of ER and PR. We have slightly modified the figure to clarify this, please find it attached as a pdf, but should you require a higher resolution jpg image do not hesitate to contact us.
We have also modified the sentence in order to assess that the combination of the low levels of ER/PR along with paracrine effects of surrounding cells might be one of the underlying mechanisms of tumor growth in myometrial stem cells (page 7, lines 260-264):
“Indeed, cells with stem or progenitor characteristics can be isolated from myometrial and leiomyoma tissues [136,137]. Interestingly, these cells seem to have low levels of sex steroid hormone receptors but require these steroids for tumor growth, suggesting that an additional paracrine mechanism, exerted by mature surrounding cells that express these receptors, is necessary for the transformation of these cells [138].”